# Complex Habitat Deconstruction and Low-Altitude Remote Sensing Recognition of Tobacco Cultivation on Karst Mountainous

Youyan Huang [1,2], Lihui Yan [1,2,*], Zhongfa Zhou [1,2,3], Denghong Huang [1,2], Qianxia Li [2,3], Fuxianmei Zhang [1,2] and Lu Cai [1,2]

[1] School of Karst Science, Guizhou Normal University/State Engineering Technology Institute for Karst Desertification Control, Guiyang 550003, China; 21010170566@gznu.edu.cn (Y.H.); fa6897@gznu.edu.cn (Z.Z.); hdh@gznu.edu.cn (D.H.); 21010170569@gznu.edu.cn (F.Z.); 232100170573@gznu.edu.cn (L.C.)

[2] Cultivation Base of State Key Laboratory of Karst Mountain Ecological Environment, Guiyang 550001, China; 20010170524@gznu.edu.cn

[3] School of Geography and Environmental Sciences, Guizhou Normal University, Guiyang 550003, China

* Correspondence: yanlihui81@163.com

**Abstract:** Rapidly and accurately extracting tobacco plant information can facilitate tobacco planting management, precise fertilization, and yield prediction. In the karst mountainous of southern China, tobacco plant identification is affected by large ground undulations, fragmented planting areas, complex and diverse habitats, and uneven plant growth. This study took a tobacco planting area in Guizhou Province as the research object and used DJI UAVs to collect UAV visible light images. Considering plot fragmentation, plant size, presence of weeds, and shadow masking, this area was classified into eight habitats. The U-Net model was trained using different habitat datasets. The results show that (1) the overall precision, recall, F1-score, and Intersection over Union (IOU) of tobacco plant information extraction were 0.68, 0.85, 0.75, and 0.60, respectively. (2) The precision was the highest for the subsurface-fragmented and weed-free habitat and the lowest for the smooth-tectonics and weed-infested habitat. (3) The weed-infested habitat with smaller tobacco plants can blur images, reducing the plant-identification accuracy. This study verified the feasibility of the U-Net model for tobacco single-plant identification in complex habitats. Decomposing complex habitats to establish the sample set method is a new attempt to improve crop identification in complex habitats in karst mountainous areas.

**Keywords:** U-Net model; complex habitat; plant identification; UAV remote sensing; karst mountainous

## 1. Introduction

Tobacco is one of the most important economic crops worldwide, mainly produced in China, the United States, India, and Brazil [1,2]. Tobacco is important in China's national economy, with more planting areas. Guizhou is an important province for tobacco planting. Tobacco has a long production cycle, high planting-labor intensity, and high technical requirements. Tobacco yield is closely associated with the survival rate of tobacco seedlings after transplanting. Obtaining accurate tobacco planting information is significant for the growth of tobacco seedlings after transplanting, tobacco fertilization, and field management [3]. Currently, tobacco seedling counting mainly relies on manual labor, which is time-consuming and labor-intensive. With the rapid development of unmanned aerial vehicles (UAV) in terms of being lightweight and stable, UAV remote sensing technology has been widely used in crop plant protection, fertilization, and growth monitoring [4–6]. Using UAV remote sensing data to identify tobacco plants and monitor plant growth information based on deep learning can save manpower and material resources and provide accurate information for large-scale growth monitoring, fertilization, and transplanting [7]. This is applicable to the management of high-value-added economic crop cultivation [8,9].

Guizhou is located in the central hinterland of one of the three major global karst regions, the southwest China karst region. This region also has the most typical karst landscapes worldwide, accounting for 62% of the total national land area. A total of 92.5% of Guizhou Province is mountainous and hilly [10]. Guizhou Province, the only province in China without the support of plains, belongs to one of the regions with the most significant karst landscape development in southwest China [11]. Due to its topography and tectonics, this province has high mountains, deep valleys, and a fragmented surface. It is affected by cloudy, rainy, and foggy weather and environmental differences. Consequently, it is difficult to obtain low and medium-resolution satellite imagery data. The information on agricultural conditions cannot be rapidly and efficiently acquired, failing to satisfy the need for agricultural monitoring [12]. Using a UAV low-altitude remote sensing platform to obtain data has the advantage of low cost and high security, mobility, and customizability. This can also effectively overcome the defect that satellite remote sensing cannot timely obtain high spatial resolution images. Thus, point–surface fusion can facilitate real-time, macroscopic, and accurate monitoring and assessment of crop growth situation can be performed through point–surface fusion in order to formulate appropriate production and management measures according to local conditions to improve crop quality and yield [13]. UAVs have the advantages of strong band continuity, large amounts of spectral data, high centimeter-level resolution, and the ability to reach areas of interest in a short period. With the increasing maturity of UAV technology, UAV multi-spectral remote sensing has been widely used for crop growth monitoring in agriculture. This facilitates easier and faster Earth observation and monitoring [14–16].

Obtaining accurate tobacco plant information in karst mountainous areas is challenging [17]. Deep neural networks were proposed in 2006 and became a popular machine learning method [18]. Due to their robustness, deep neural networks have an impressive track record of applications in image analysis and interpretation [19], initially in biomedicine and later in agriculture [20,21]. Compared with traditional methods such as a support vector machine [22,23], color space [24], random forest [25], artificial neural network (ANN) [26,27], and hyperpixel space [28], deep learning methods can overcome their shortcomings such as higher requirements for observer experience, higher labor intensity, and insufficient extraction accuracy for precision agriculture. Chen et al. [29] applied deep neural networks to high-resolution images in order to identify strawberry yield, with an average accuracy of 0.83 and 0.72 in identifying 2 m and 3 m aerial height, respectively. Oh et al. [30] used deep learning target detection technology with UAV images for cotton seedling counting and analyzed plant density and precision management. The target detection network identification method showed higher accuracy than traditional methods. Wu et al. [31] used deep learning to extract apple tree canopy information from remote images. This remote sensing technique had a precision of 91.1% and a recall of 94.1% for apple tree detection and counting, an overall precision of 97.1% for branch segmentation, and an overall precision of more than 92% for canopy parameter estimation. Deep learning methods can achieve higher accuracy than traditional methods. As the depth of deep learning models continues to increase, their feature representation ability and segmentation accuracy become increasingly higher. Despite these advantages, there are some shortcomings. Deeper models are more complex and require more training samples, higher hardware and software configuration for operation and longer training time [32].

However, the U-Net model can overcome these shortcomings. Freudenberg et al. [33] used the U-Net neural network to identify palm satellite image maps with a resolution of 40 cm and found that the method was reliable even in shaded or urban areas, with palm identification accuracy ranging from 89% to 92%. Yang et al. [34] used the FCN-AlexNet and SegNet models to estimate the rice fall area in UAV imagery, and the F1-score reached 0.80 and 0.79, respectively. Using cigar tobacco plants as the research object, Rao et al. [35] proposed a new deep learning model to learn the morphological features of the center of the tobacco plants through some key features. They adopted a lightweight coder and decoder to rapidly identify the tobacco and locate the counts from UAV remote sensing imagery,

with an average accuracy of up to 99.6%. Li et al. [36] extracted dragon fruit plants from UAV visible images of different complex habitat strains based on the U-Net model. The identification accuracies were 85.06%, 98.83%, and 99.20% for the initial, supplementary and extended datasets, respectively. Their experimental results show that increasing the type and number of samples can improve the model's accuracy in identifying dragon fruit plants, and the accuracy of the U-Net model was also verified. The applicability of the U-Net network model was verified in identifying features in plateau mountainous areas. Huang et al. [37] proposed an accurate extraction method of flue-cured tobacco planting areas based on a deep semantic segmentation model for UAV remote sensing images of plateau mountainous areas. A total of 71 scene recognition images were semantically segmented using DeeplabV3+, PSPNet, SegNet, and U-Net, with segmentation accuracies of 0.9436, 0.9118, 0.9392, and 0.9473, respectively. Deep learning-based methods can overcome the problem of insufficient characterization ability of traditional machine vision methods. However, they need a large amount of sample data for training [38]. Under deeper model layers, these models also require longer training time [39]. In contrast, the U-Net model can obtain higher recognition results with fewer training samples. This model needs less training time relative to other models, such as convolutional neural networks (CNN) [40] and fully convolutional networks (FCN) [41], and saves experiment time with higher running speed. Due to the complexity of the tobacco-planting environment, it is difficult for traditional methods to extract high-precision tobacco plant information from UAV images. Thus, finding a new method to decompose complex scenes into multiple homogeneous scenes and then perform scene-by-scene identification to improve the overall accuracy is necessary.

In summary, most of the existing studies have extracted information on tobacco cultivation, and many models used in existing research have mainly been used for crops with relatively simple growth conditions and plant types, strong regularity of sowing spacing and plant types, relatively uniform spatial distribution of crop plants, relatively simple growing environmental conditions, and obvious and relatively homogeneous features of crop remote sensing images. In contrast, crops in karst mountainous areas have complex growing environments and structures. The crops show a significantly dispersed three-dimensional spatial distribution and have nonuniform plant sizes. This study used the U-Net model as a binary semantic segmentation method for UAV visible light images of tobacco plants at the root extension stage in complex habitats. According to the tobacco-planting environment in the study area, the complex habitat was divided into eight tobacco plant recognition habitats by considering four main factors (i.e., plot fragmentation, plant size, presence of weeds, and shadow masking). The accuracy of each scenario was evaluated to analyze the influencing factors of the recognition accuracy. This can promote the application of UAV remote sensing in agriculture, accelerate dataset standardization, and provide data and methodological support for fine agricultural management in the karst mountainous areas.

## 2. Materials and Methods

### 2.1. Study Area

The study area (Figure 1) is located in Beipanjiang Town, Zhenfeng County, Qianxinan Prefecture, Guizhou Province (105°35′53″ E, 25°36′08″ N). Beipanjiang Town is a karst geomorphological area with a rugged and fragmented surface. The terrain in the territory is high in the south, low in the north, hilly in the northeast, and smooth in the center, with complicated topography and a relative altitude of 1475 m. The Beipanjiang River Valley slope area has a deep cut. The climate is characterized as the subtropical monsoon humid climate, with four distinct seasons. The annual average mild frost-free period, sunshine hours, number of precipitation days, and precipitation are 300 days, 1549.2 h, 180 days, and 1100 mm, respectively. Due to the mild summer and winter, concurrent rain and heat are the most suitable for tobacco planting. However, the study area has mountainous characteristics such as fragmented cropland, the coexistence of regular and

narrow croplands, the coexistence of clear-contour and fuzzy-boundary croplands, cropland patches with high fragmentation, and diverse farming methods. In 2023, more than 6900 acres of tobacco were planted in Beipanjiang Town, which is expected to achieve an output value of more than CNY 25 million.

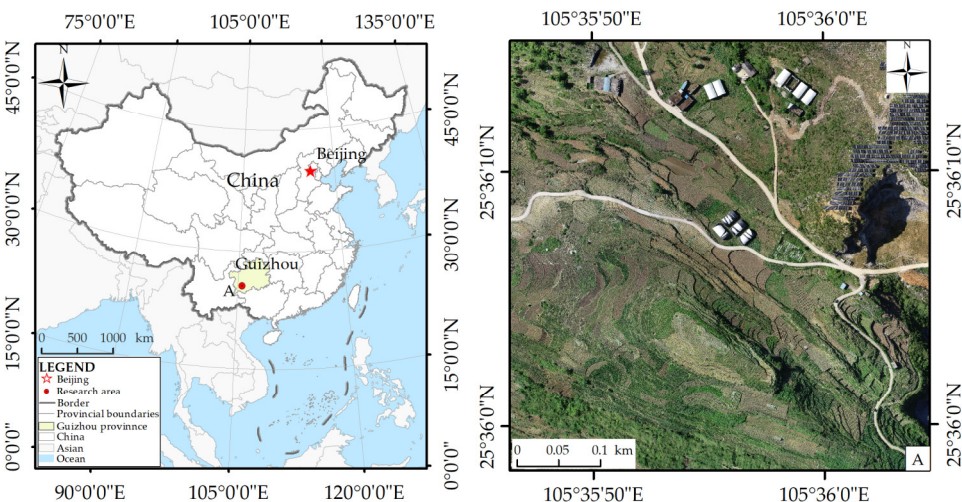

**Figure 1.** Location maps of the study area: UAV image of the study area.

## 2.2. Data Acquisition and Preprocessing

Due to rainy and cloudy weather and rugged and fragmented terrain, it is difficult to acquire satellite optical remote sensing data in karst mountainous areas. Therefore, this study used the UAV DJI (DJ-Innovations, acronyms DJI, Shenzhen,China) Mavic2 Pro v2.0 as the image data-acquisition platform. This platform was equipped with a 1-inch CMOS sensor Hasselblad camera with 20 million photo pixels, a resolution of 5472 × 3684 pixels, and a maximum wind resistance level of 5. It is small, low-cost, mobile, and flexible and does not require a wide-level site for take-off and landing. Thus, it is suitable for collecting data in mountainous environments with steep terrain, fragmented land, and difficulties in obtaining high-precision satellite image data. The image was acquired between 15:00 and 16:00 on 4 June 2021, under clear weather and wind force 2.5, meeting safe UAV operation requirements. In order to ensure the accuracy and quality of remote sensing images during flight, the UAV captured images in the waypoint flight mode, with a heading overlap rate of 80%, a side overlap rate of 75%, and a flight altitude of 120 m. This can facilitate clear images with good quality.

The UAV photos were processed using Pix4Dmapper4.0 software for initialization, feature point matching, image stitching, correction (deformation, distortion, blurring, and noise due to UAV shaking), image enhancement, color smoothing, cropping, and reconstruction to generate a high-resolution orthophoto map (Digital Orthophoto Map, DOM). Finally, orthophotos with a spatial resolution of 6.4 cm were obtained.

## 2.3. Network Modeling and Model Parameter Selection

### 2.3.1. U-Net Model

The U-Net model is a network structure based on CNN proposed by Ronneberger et al. in 2015 [42]. This model was initially applied to the semantic segmentation of medical images and achieved good performance in different biomedical segmentation applications [43]. Then, the U-Net network model was applied to agriculture. In recent years, the U-Net model has made great progress in agricultural remote sensing and crop recognition [44–48]. Its structure is shown in Figure 2, consisting of the compressing path in the left half and the expansive path in the right half. The core idea of the model is the introduction of jump connections, greatly improving the accuracy of image segmentation. In contrast to CNNs, U-Net uses feature splicing to achieve feature fusion [49]. Due to the elastic deformation of

data enhancement [50], encoders usually superimpose convolution and pooling operations to gradually reduce the size of feature maps. Consequently, a large number of parameters are introduced, reducing the model's efficiency. In addition, downsampling constantly loses spatial information. This can result in the loss of the image's deep details and affect the final segmentation result [51]. However, in contrast to other deep learning models, the U-Net model is also suitable for rapid crop extraction due to its advantages of "fewer training samples, shorter training time and higher accuracy".

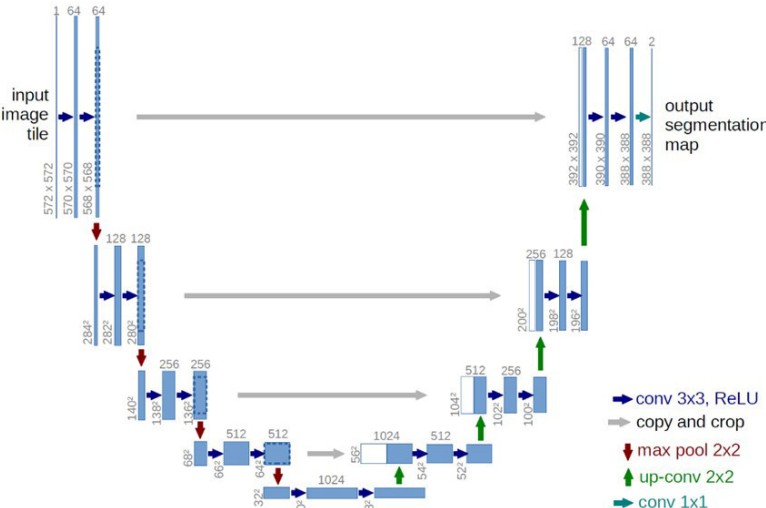

**Figure 2.** The U-Net model [4].

### 2.3.2. Experimental Environment

The experimental study was conducted on a professional imaging workstation equipped with Windows 10 (ACPIx64 processor). The computer was powered by an NVIDIA GeForce RTX 2080 Ti (GPU) and an Intel(R) Core(TM) i9-10980XE CPU to accelerate relevant operations. The study was based on the Tensorflow-GPU version 2.0.0 deep learning framework and used the Adma optimizer as the optimization function. Keras = 2.4.3 is a WrapperAPI of Tensorflow, a layer of Tensorflow wrapping that allows for simpler model building [36]. The initial learning rate was set as 0.0001 for the model training. The total number of iterations was 50. The training was performed on the workstation. The model was constantly debugged to obtain optimal parameters, improving the recognition accuracy of the U-Net model.

### 2.3.3. Model Parameter Selection

In order to study the influencing factors of complex tobacco habitats in the karst mountainous area on identifying extracted tobacco using the U-Net model, two groups of training samples were preset for model training. The first set of the experiments was to train the model with all the training samples and labels, and the second set was to train the model with eight habitats: smooth tectonics and weed-free (I); smooth tectonics and unevenly growing (II); smooth tectonics and weed-infested (III); smooth tectonics and planted with smaller seedlings (IV); subsurface fragmented and weed-free (V); surface fragmented and shadow-masked (VI); subsurface fragmented and weed-infested (VII); and surface fragmented and planted with smaller seedlings (VIII).

In order to obtain the optimal tobacco identification model parameters, multiple parameters can be set to compare the model training. Figure 3 presents the accuracy and loss-changing curves of the model trained with different parameters. The parameter changes included the learning rate and the number of iterations. There were some differences in the trend of the loss and accuracy curves of the model trained with different parameters. In order to explore more suitable model parameters for tobacco identification, all the samples and labels of eight habitats were used for model training together, with a

ratio of 8:2 for the training set and the test set. Firstly, the number of iterations was set as 50, and the learning rate was 0.0001 (Figure 3a,b) and 0.001 (Figure 3c,d), respectively. It can be concluded from the experiments that the learning rate of 0.0001 was more suitable for the tobacco identification model. Secondly, the learning rate was set as 0.0001, and the number of iterations was set as 100 and 50. When the number of iterations was set as 50, the loss and the accuracy curves in Figure 3e,f showed better fitting results. The loss and accuracy curves under 100 iterations are shown in Figure 3g,h. After many rounds of model training, the comparative analysis shows that the model was more robust when the learning rate and the number of iterations were set as 0.0001 and 50, respectively. Therefore, the model parameter setting with a learning rate of 0.0001 and 50 iterations in the whole model training can meet the research needs.

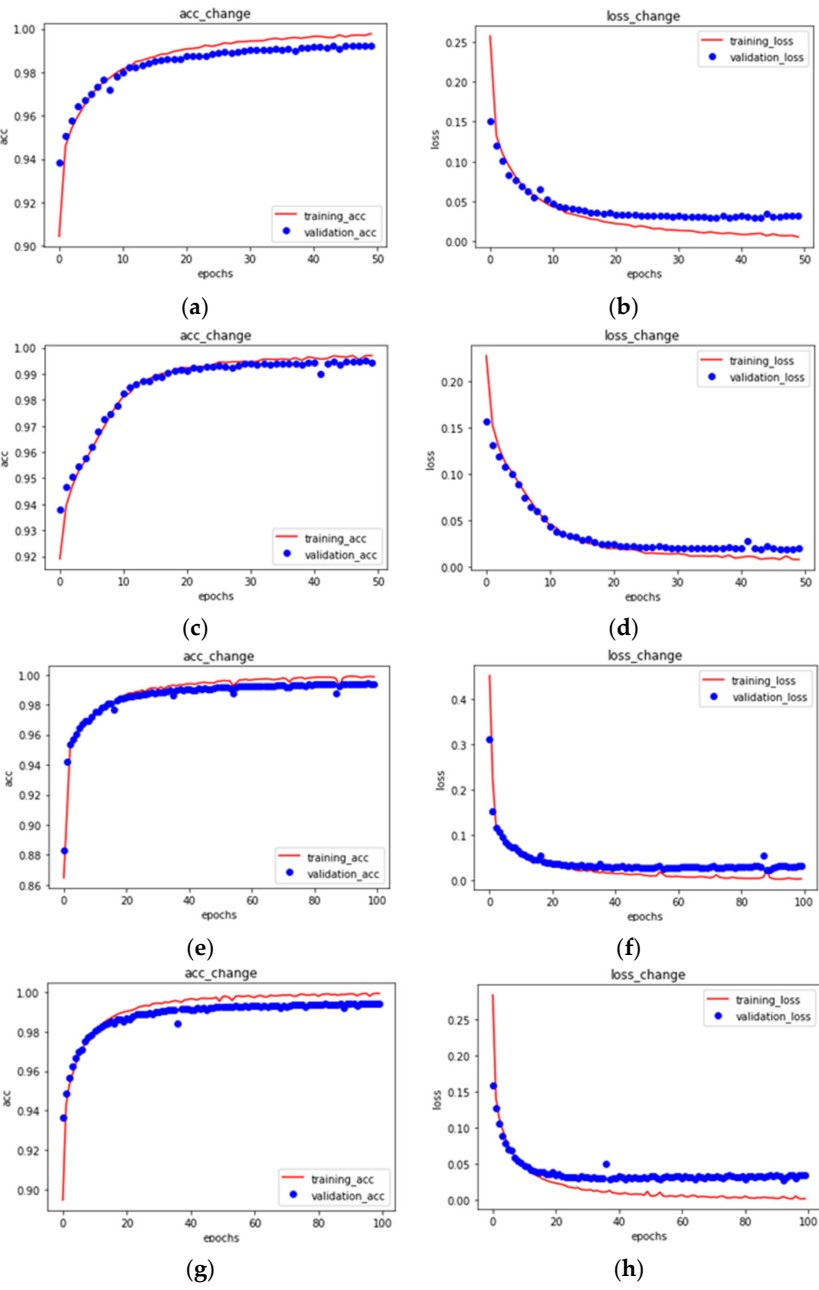

**Figure 3.** U-Net modeling accuracy and loss curves. ((**a**,**b**) are the learning rate of 0.0001 and the epochs of 50 the model training curves; (**c**,**d**) are the learning rate of 0.001 and the epochs of 50 the model training curves; (**e**,**f**) are the learning rate of 0.0001 and the epochs of 100 the model training curves; (**g**,**h**) are the learning rate of 0.001 and the epochs of 100 the model training curves).

### *2.4. Dataset Construction*

2.4.1. Classification of Complex Habitat for Tobacco

The dataset used for model training is also known as the training sample, which is the basis of the whole model classification algorithm. The quality of training samples directly affects classification results. Therefore, representative and typical samples with the completeness of regional sample points should be selected [52]. In order to better extract information on complex habitats of tobacco, the UAV visible light images in June 2021 were selected to extract tobacco plants according to the complexity of the tobacco-planting habitat in the study area. The tobacco plants were at the rooting stage, with nonuniform growth and size. In order to better analyze the model recognition accuracy under the complex crop growth habitat and explore the suitability of different habitats, four main factors were considered, i.e., the plot fragmentation, plant size, presence of weeds, and shadow masking according to the tobacco planting habitat in the study area. Then, the eight habitats were classified, as shown in Figure 4: smooth tectonics and weed-free (I); smooth tectonics and unevenly growing (II); smooth tectonics and weed-infested (III); smooth tectonics and planted with smaller seedlings (IV); subsurface fragmented and weed-free (V); surface fragmented and shadow-masked (VI); subsurface fragmented and weed-infested (VII); and surface fragmented and planted with smaller seedlings (VIII). In this scene classification system, training samples were constructed based on the UAV visible light images.

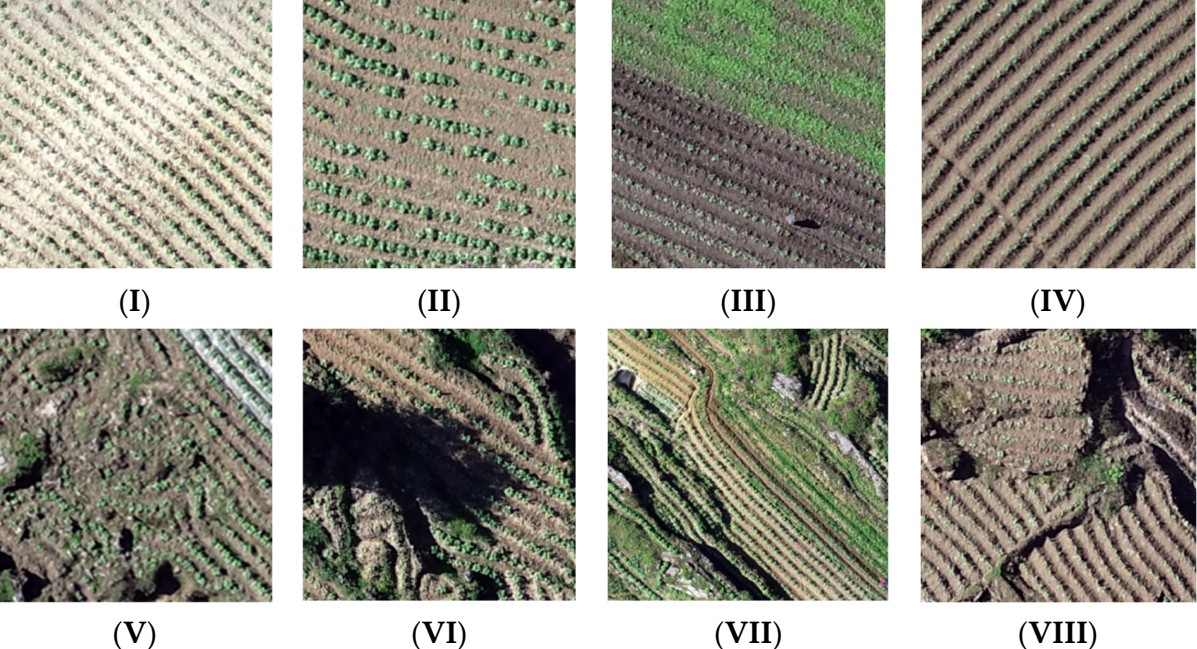

**Figure 4.** Study area habitat delineation map.(smooth tectonics and weed-free (**I**); smooth tectonics and unevenly growing (**II**); smooth tectonics and weed-infested (**III**); smooth tectonics and planted with smaller seedlings (**IV**); subsurface fragmented and weed-free (**V**); surface fragmented and shadow-masked (**VI**); subsurface fragmented and weed-infested (**VII**); and surface fragmented and planted with smaller seedlings (**VIII**)).

### 2.4.2. Construction of Sample Datasets

The ROI tool of ENVI5.3 (Exelis Visual Information Solutions, Dallas, TX, USA) was used to manually annotate the outline of the tobacco plants in order to generate .xml files. These files were then converted to vector files. The pixels of tobacco plants were labeled as black (pixel value was 0), and the pixels of non-tobacco plants were labeled as white (pixel value was 255) using the ArcGIS 10.2 (ESRI, Redlands, CA, USA) conversion tools. The binary mapping of labels was used to evaluate the segmentation and the information extraction of the tobacco plants. A total of 6617 plants were labeled. Since the whole UAV image was directly used as a sample, the data volume was too large. Thus, the performance requirements for the computer were high. This was not conducive to model training. Thus, the images and the corresponding labeled tobacco plants were randomly cut into samples with a size of 224 × 224 pixels. The randomly cut samples have cross-overlapping parts with random sizes, inducing different samples and enhancing the randomness of the samples. The information in the UAV visible light images can be fully utilized. Finally, 2300 samples were obtained to constitute the tobacco dataset. The tobacco dataset included a sample image folder and a manually labeled label folder. The sample image folder corresponded to the sample images, which were named and arranged in numerical order. The label file contained the manually labeled data corresponding to the sample images. Some captured tobacco plant images and corresponding labels are shown in Figure 5.

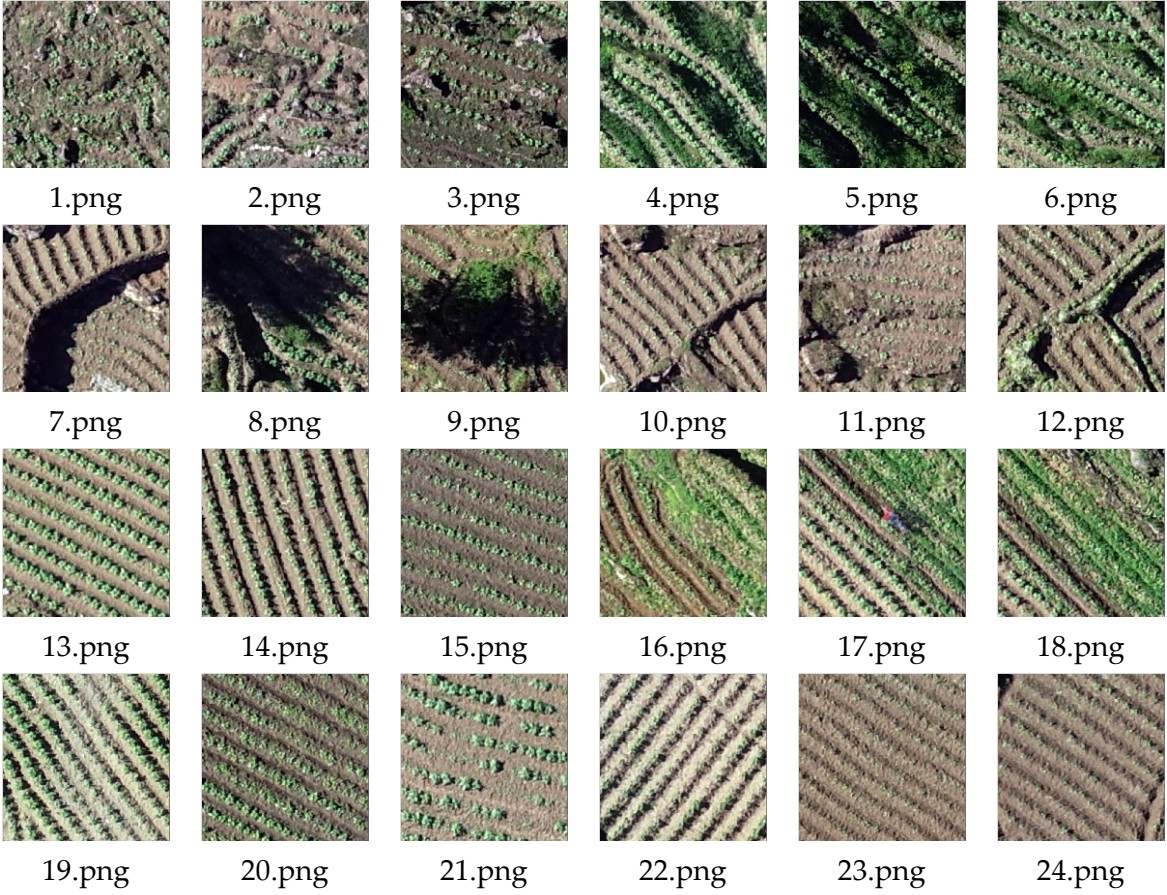

**Figure 5.** *Cont.*

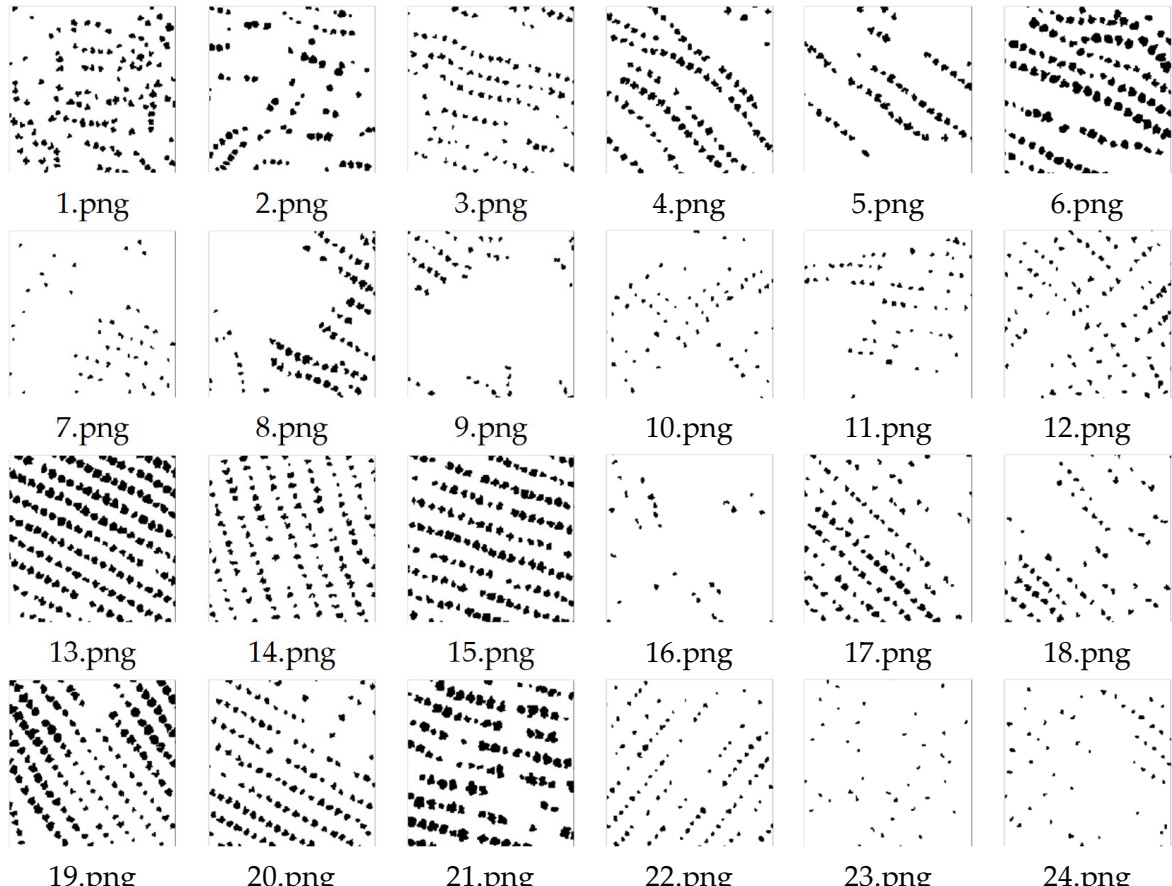

**Figure 5.** Images of tobacco with manual annotation labels: subsurface fragmented and weed-free (1.png, 2.png, and 3.png); subsurface fragmented and weed-infested (4.png, 5.png, and 6.png); surface fragmented and shadow-masked (7.png, 8.png, and 9.png); surface fragmented and planted with smaller seedlings (10.png, 11.png, and 12.png); smooth tectonics and weed-free (13.png, 14.png, and 15.png); smooth tectonic and weed-infested (16.png, 17.png, and 18.png); smooth tectonics and unevenly growing (19.png, 20.png, and 21.png); smooth tectonics and planted with smaller seedlings (22.png, 23.png, and 24.png).

2.4.3. Optimization Sample

In order to improve the generalization ability and segmentation accuracy of the U-Net model, the samples in the datasets were optimized. The optimized datasets can provide samples closer to real tobacco plants (such as geometric morphological features) for model training. Particularly, some samples were interfered with by environmental factors such as light, shadow, and ground reflection. To accommodate these interferences, the effects of sample quantity and quality on model accuracy were explored. The ArcGIS10.2 software (https://www.esri.com/en-us/home, accessed on 12 January 2024) was used to optimize the sample dataset and enhance the tobacco plants. The following four aspects were mainly addressed and improved: (1) boundary accuracy for samples with smaller plants; (2) difficulties in distinguishing tobacco plants accompanied by weeds; (3) weak ground reflections and shadow interference; the interference of incorrectly labeling shadows as tobacco samples can be excluded; and (4) strong ground reflection and indistinct tobacco plant information characteristics; consequently, tobacco plant samples are missed. Partial samples and optimized samples are shown in Figure 6, where the blue color indicates original samples and the yellow color indicates optimized samples. In addition, since the samples were randomly split, the number of training samples was increased to 9500 in order to enrich the content and diversity of the samples.

| Typology | **Samplel1** | **Samplel2** | **Samplel3** | **Samplel4** |
|---|---|---|---|---|

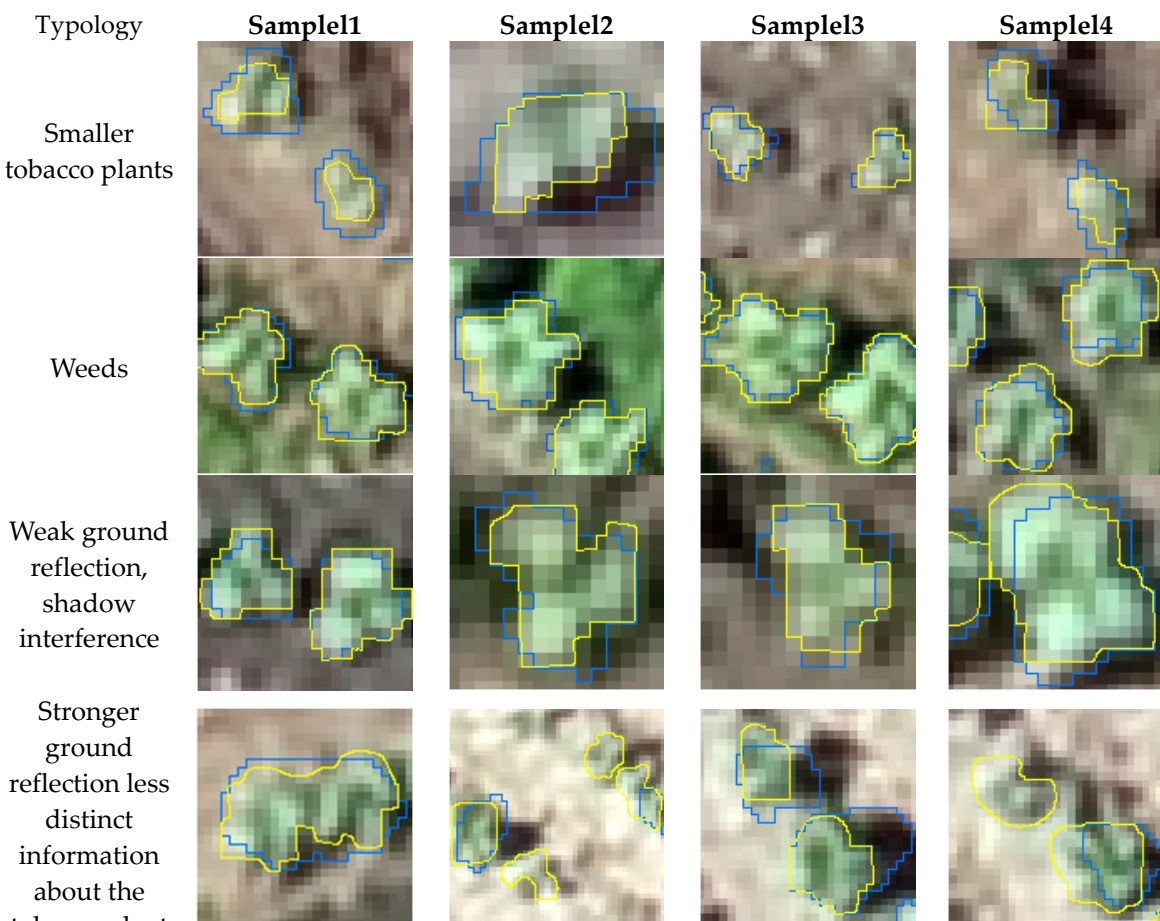

Smaller tobacco plants

Weeds

Weak ground reflection, shadow interference

Stronger ground reflection less distinct information about the tobacco plant

**Figure 6.** Comparison between partial samples and optimized samples. (tips:where the blue color indicates original samples and the yellow color indicates opti-mized samples).

*2.5. Evaluation Index*

Multiple metrics are usually used in image segmentation to evaluate algorithm precision. In this study, four quantitative metrics, namely, precision, recall, F1-score, and Intersection over Union (IOU), were used to quantitatively evaluate the model recognition results and the segmentation precision of the tobacco plants in each scene of the UAV remote sensing images.

Precision indicates the probability of actually being a positive sample out of all samples identified as positive:

$$\text{precision} = \frac{\text{TP}}{\text{TP} + \text{FP}} \tag{1}$$

Recall is used to find how many samples that are actually positive are identified as positive:

$$\text{Recall} = \frac{\text{TP}}{\text{TP} + \text{FN}} \tag{2}$$

The F1-score is a common measure for classification problems and is a harmonic mean of precision and recall ranging from 0 to 1. The closer the F1-score is to 1, the more robust the model is:

$$\text{F1-score} = \frac{2}{\frac{1}{\text{Precise}} + \frac{1}{\text{Recall}}} = 2 \times \frac{\text{Precise} \times \text{Recall}}{\text{Precise} + \text{Recall}} \tag{3}$$

IOU is a commonly used evaluation method in semantic segmentation and can measure the degree of overlap between the target detection frame and the true frame. Thus, IOU can be used as a criterion to determine whether the detection frame is a positive sample

or not. Comparison with the threshold can help to determine whether it is a positive or negative sample. Generally, when the identified frame and the real frame IOU $\geq$ 0.5, it is considered to be a positive sample:

$$IOU = \frac{TP}{TP + FP + FN} \qquad (4)$$

where TPindicates that a tobacco sample is correctly identified as tobacco, FN indicates that a tobacco sample is incorrectly identified as non-tobacco, and FN indicates that a non-tobacco sample is incorrectly identified as tobacco.

## 3. Results

### 3.1. Quantitative Analysis of Plant Extraction Precision

Using the U-Net model to identify UAV remote sensing visible flue-cured tobacco images, the accuracy of the segmentation results is shown in Table 1. The recognition accuracies of the eight habitats were in the following order: subsurface fragmented and weed-free (V) > surface fragmented and planted with smaller seedlings (VIII) > subsurface fragmented and weed-infested (VII) > smooth tectonics and weed-free (I) > smooth tectonics and unevenly growing (II) > surface fragmented and shadow-masked (VI) > smooth tectonics and planted with smaller seedlings (IV) > smooth tectonics and weed-infested (III). Comparing the whole image with the recognition results of the eight scenes, Scenes III and IV showed lower accuracy. Then, the other scenes were compared with the whole image of the study area, and the overall accuracy of the whole image was lower, with a precision of 0.68, a recall of 0.85, an F1-score of 0.75, and an IOU of 0.60. Maize, as the same green crop as tobacco, was incorrectly identified as tobacco by the U-Net model. This is due to the fragmented surface, complex planting structure, mixed cultivation of tobacco and maize plots, and more bushes and weeds along the cultivated soil canals. Thus, the accuracy of the whole image was low.

**Table 1.** Identification results for different scenes.

| Scenes | Precision | Recall | F1-Score | IOU |
|---|---|---|---|---|
| Smooth tectonics and weed-free (I) | 0.76 | 0.86 | 0.81 | 0.67 |
| Smooth tectonics and unevenly growing (II) | 0.74 | 0.89 | 0.81 | 0.67 |
| Smooth tectonics and weed-infested (III) | 0.49 | 0.69 | 0.57 | 0.40 |
| Smooth tectonics and planted with smaller seedlings (IV) | 0.58 | 0.79 | 0.67 | 0.50 |
| Subsurface fragmented and weed-free (V) | 0.85 | 0.84 | 0.84 | 0.73 |
| Surface fragmented and shadow-masked (VI) | 0.73 | 0.87 | 0.79 | 0.66 |
| Subsurface fragmented and weed-infested (VII) | 0.77 | 0.88 | 0.82 | 0.69 |
| Surface fragmented and planted with smaller seedlings (VIII) | 0.77 | 0.79 | 0.78 | 0.64 |
| The whole image | 0.68 | 0.85 | 0.75 | 0.60 |

Subsection-scene-recognition accuracy is shown in Figure 7, with some differences. The factors affecting the recognition accuracy in each scene were also different. The scene of "subsurface fragmented and weed-free (V)" had the highest accuracy (precision = 0.85), followed by "surface fragmented and planted with smaller seedlings (VIII)" (precision = 0.77). The scene of "smooth tectonics and weed-infested (III)" had the lowest recognition accuracy (precision = 0.49).

The tobacco plants were smaller and fuzzy in the training samples. The contours of the tobacco were unclear. The model would omit smaller tobacco plants, leading to a lower accuracy in recognizing Scene IV.

Comparative analysis reveals that the recognition accuracy of Scene VIII was higher than that of Scene IV. This is mainly attributed to the relatively homogeneous planting habitat in Scene VIII, with more bare rock on the fragmented surface, fewer green vegetation such as weeds, and significant differences between tobacco plants and background texture

features in the images. Thus, the U-Net model was less affected in tobacco identification, resulting in higher identification accuracy of tobacco plants.

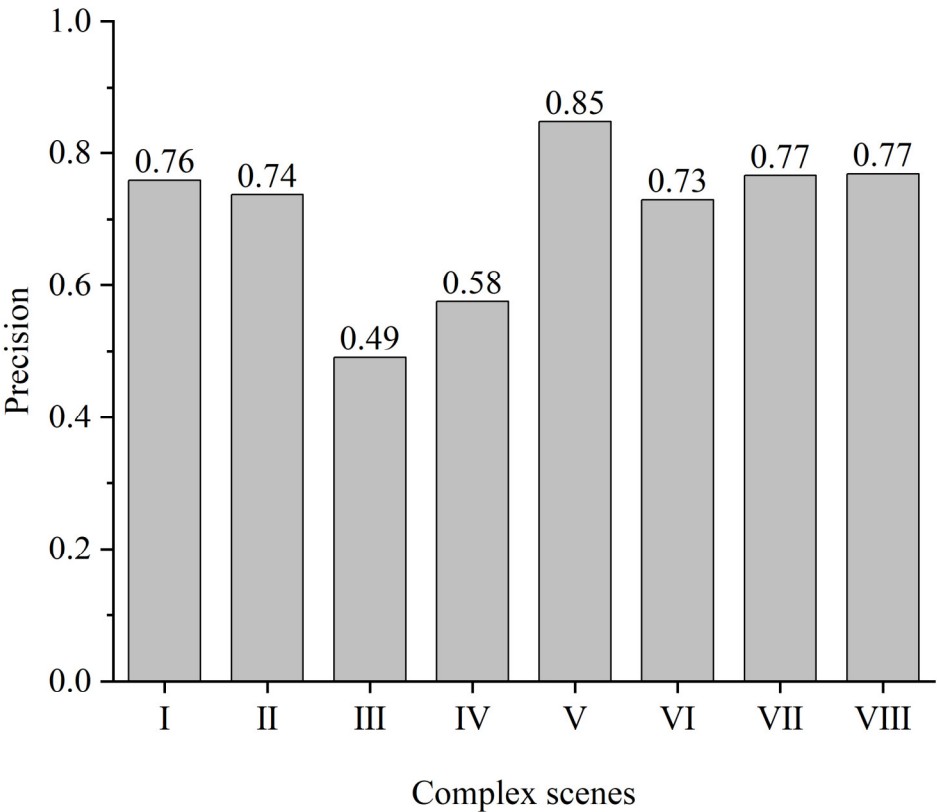

**Figure 7.** Complex identification precision histogram.

### 3.2. Visual Analysis of Tobacco Plant Extraction

In Figure 8, based on the model, sample quality and quantity, and complex habitat dataset recognition results, the red contour represents the identified tobacco plants. Scenes I and V were taken as the control. Scene I had regular tobacco planting and connected tobacco leaves, with overall even recognition accuracy. Scene V had irregular tobacco planting. Tobacco plants were smaller due to the water and fertilizer conditions. Its single-plant recognition accuracy was high.

Scene III had regular tobacco planting and good plant growth. The model identified non-tobacco parts as tobacco plants during continuous plant identification. Weeds and tobacco plants are both green vegetation and have similar spectral and texture features. These were the main reasons for the low recognition accuracy of Scene III. Scene VII had irregular tobacco planting. There was weed confusion to some extent. Compared with Scene III, Scene VII had less continuous tobacco planting. Thus, the recognition accuracy of Scene VII increased by 0.28 compared to that of Scene III.

Scenes IV and VIII were taken as the control. Tobacco plants in Scene IV were affected by the transplanting time sequence, and the tobacco plant seedlings were smaller than those in Scene VIII. Thus, the tobacco feature information of the training samples was insignificant. The U-Net model may omit smaller tobacco plants during training, resulting in a lower identification accuracy of Scene IV than that of Scene VIII by 0.19.

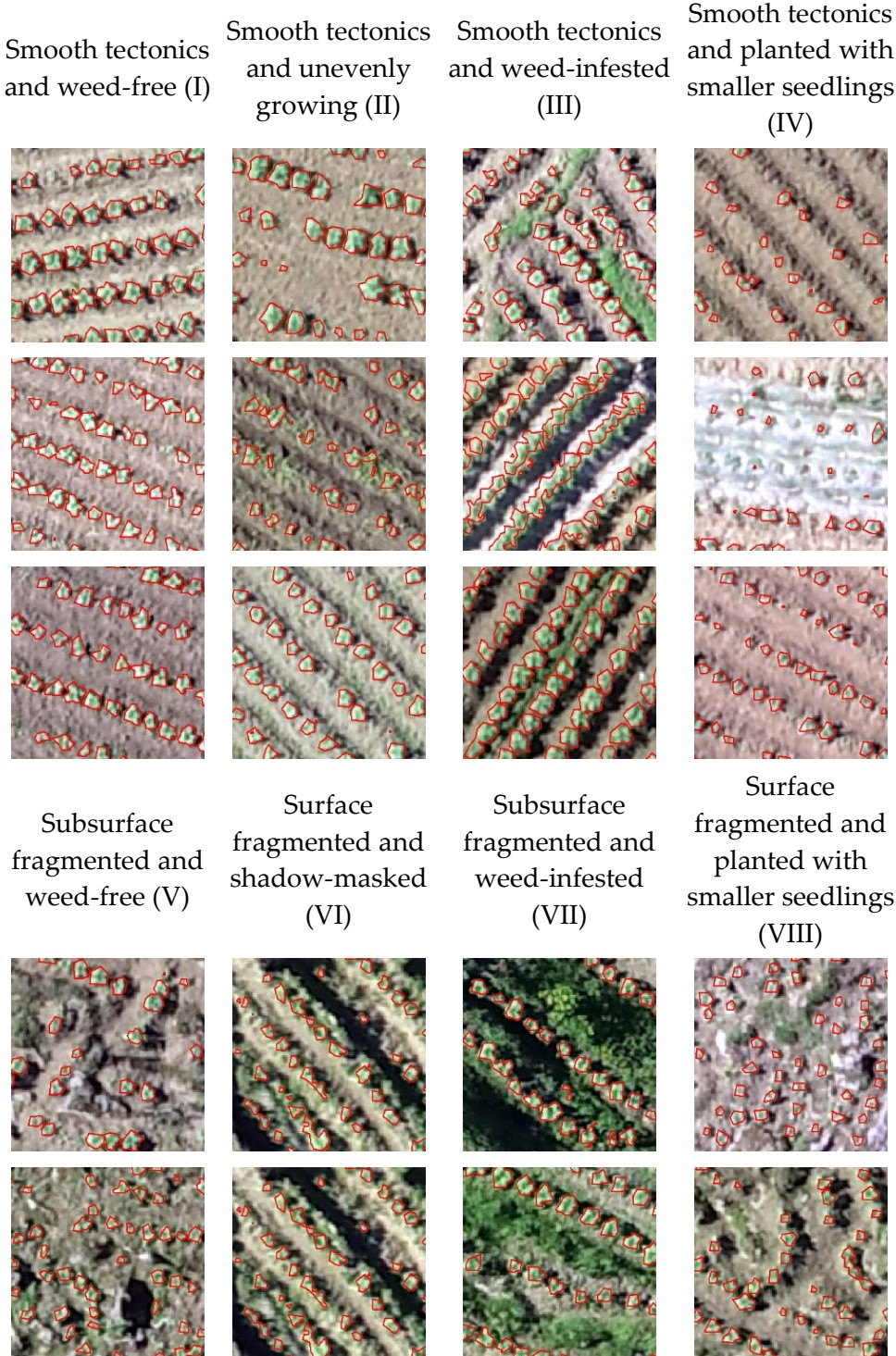

**Figure 8.** Identification results of the U-Net model. (the red outlines in the figure indicate the tobacco plant outlines recognized by the U-Net model).

*3.3. Optimization Sample Precision Analysis*

The U-Net model was trained using the optimized sample dataset. The results show that the recognition accuracy of the eight complex scenes was generally improved, as shown in Figure 9. Among them, the scene broken surface with weeds (VII) had the highest recognition accuracy, followed by broken surface shadow masking (VI) with 0.92 and 0.90, respectively. The optimized samples were targeted at the scene with broken surfaces with weeds to clarify the boundaries of tobacco and weeds in the training samples. Tobacco

plants on the broken surface were distinguished from bare rocks. The characteristics of tobacco plants were significant. The model showed strong generalization ability with high recognition accuracy. In the fragmented surface and shadow masking (VI), the area without tobacco blocked by trees was regarded as a non-tobacco area. The UAV images were combined with the analysis of synthetic images to improve the recognition accuracy. Scene III had the lowest accuracy rate, followed by Scene II, with accuracy rates of 0.74 and 0.78, respectively. Scene III had a relatively flat ground. The drone acquired the image in the afternoon, and the shadow of the weeds and tobacco plants had a greater impact on the recognition accuracy of the tobacco plants and reduced the model's accuracy in identifying tobacco plants.

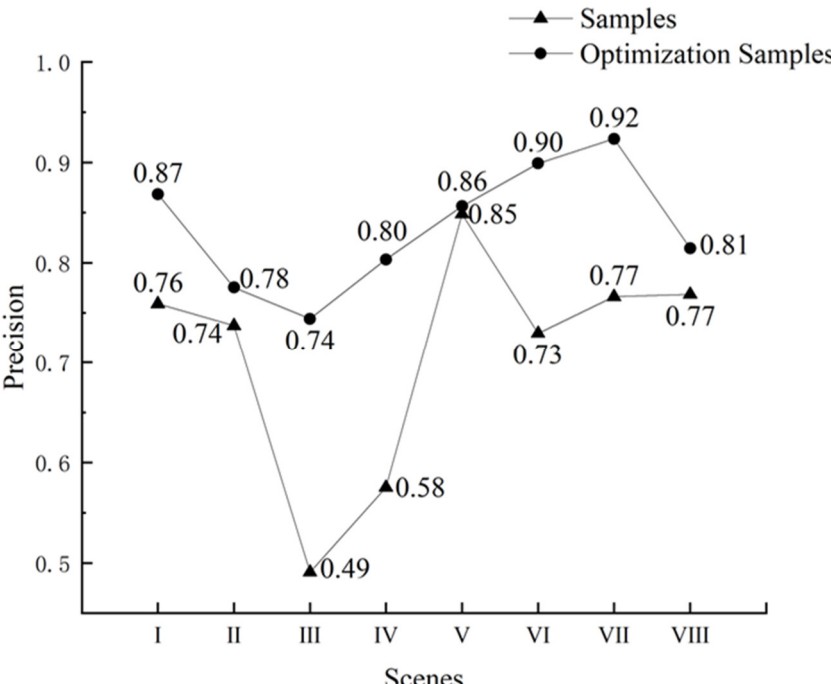

**Figure 9.** Recognition precision for the original sample and optimized sample datasets.

## 4. Discussion

To address the problems of a fragile natural environment, fragmented plots, and complex planting structures in the karst mountain areas of southern China, we constructed eight sample datasets of tobacco plants in complex scenes to train the U-Net model. Tobacco information was accurately extracted from UAV remote sensing imagery in complex scenes. The factors affecting the recognition accuracy of the U-Net model in different complex scenes were discussed in two aspects: tobacco plant omission and wrong extraction.

### 4.1. Analysis of Omitted Factors

In order to investigate the factors affecting the U-Net model on the segmentation of tobacco plants in different complex habitats, we analyzed the influencing factors through the segmentation results of the eight habitats. The segmentation accuracy of Scene III was the lowest, with a precision of 0.49 and a recall of 0.69. A total of 410 plants were omitted, including 95 whole plants (23%) and 315 incomplete plants (77%). The omission of complete plants mainly included two factors: weed cover and small saplings of tobacco plants. For incomplete tobacco plants, the omission mainly resulted from the small size of tobacco plants covered by weeds. Weeds and tobacco plants had similar texture and spectral features. The soil background had low reflectance. These factors affected the recognition accuracy of the scene with smooth tectonics and weed cover. Overall, the omission of the tobacco plant in this scene was mainly attributed to weed cover.

For tobacco images in Scene IV, 420 plants were missed, including 187 whole plants (45%) and 233 incomplete tobacco plants (55%). The omission of the whole tobacco plants was mainly because the tobacco plant sapling was smaller. Larger UAV flight height reduced the UAV image resolution, resulting in low tobacco identification accuracy. In the low-altitude remote sensing multi-scale recognition of complex habitats in karst mountainous areas, Li [4] found that the accuracy of UAV images for tobacco plant recognition decreased with increasing height. In this study, the UAV flight altitude was 120 m. The tobacco plants at the rooting stage were small. Thus, the image resolution was low, and some tobacco plant features were lost. This resulted in the low accuracy of the U-Net model in segmenting the tobacco plant scene

### 4.2. Analysis of Erroneous Factors

Further detailed analysis was conducted to reveal the factors affecting the recognition accuracy of the model incorrectly identifying non-target features as target features (tobacco). The model incorrectly identifying non-target features as target features was referred to as misidentification. The six main factors that caused misidentification were identified using overlay analysis of the experimental results (Table 2), i.e., the edge of tobacco plants, maize plants, bushes, white mulch, bare rocks, and weeds. Particularly, the edge of the tobacco plants was the most mislabeled, accounting for 67.21% of the whole image mislabeled. This is mainly because the segmentation level of the U-Net model was at the pixel level. In addition, the data was collected between 15:00 and 16:00. The sun altitude angle varied. There was a shadow on the leaves of the tobacco plants, and the model incorrectly identified the shadow of the tobacco leaves as the real tobacco leaves. Thus, there was a discrepancy between the validation labels and the manually drawn ones in identifying the contour of the tobacco plants.

The second was the misidentification caused by the confusion of maize plants and weeds, accounting for 24.50% of the misidentification of the whole image, and weeds accounted for 7.40%. The karst mountainous area had fragmented surfaces, scattered cultivated plots, and complex planting structures. The tobacco plots were adjacent to the maize plots and covered with weeds. The tobacco plants and weeds were all green vegetation with similar shapes, textures, and spectral features. Thus, the model incorrectly identified maize and weeds as tobacco, leading to more misidentification in the U-Net model and lower recognition accuracy. The U-Net model may incorrectly identify weeds as tobacco plants during tobacco identification.

**Table 2.** Statistics of factors influencing misidentification.

| Factor | Tobacco | Maize | Weeds | Shrub | Bare Rock | White Plastic |
|---|---|---|---|---|---|---|
| Typical incorrect recognition patches |  |  |  |  |  |  |
| Misidentification | 1,008,072.52 | 367,544 | 110,935 | 7643 | 3204 | 2574 |
| Percentage | 67.21% | 24.50% | 7.40% | 0.51% | 0.21% | 0.17% |

### 4.3. Analysis of the Impact of Optimized Samples on the Accuracy of the Model in Identifying Tobacco Plants

Figure 10 and Table 3 indicate that the sample quality had a certain impact on the recognition accuracy of the U-Net model. In this experiment, the recognition accuracy of the model trained by the optimized samples was higher than that of the original samples. The recognized tobacco plants were closer to the real tobacco plants. The identification accuracy of Scene III increased by 25.31%. In order to better evaluate the impact of the sample on the model, the results from the scene with smooth tectonics were used as the evaluation

indicator of the model trained by the two dataset samples. The calculation results were analyzed to reveal their differences. Positive values indicate that the accuracy is improved, and negative values indicate that the accuracy is reduced. Regarding the accuracy, Scene IV showed the highest increase (22.81%). Scene V showed the lowest increase (0.78%), followed by Scene II (3.82%).

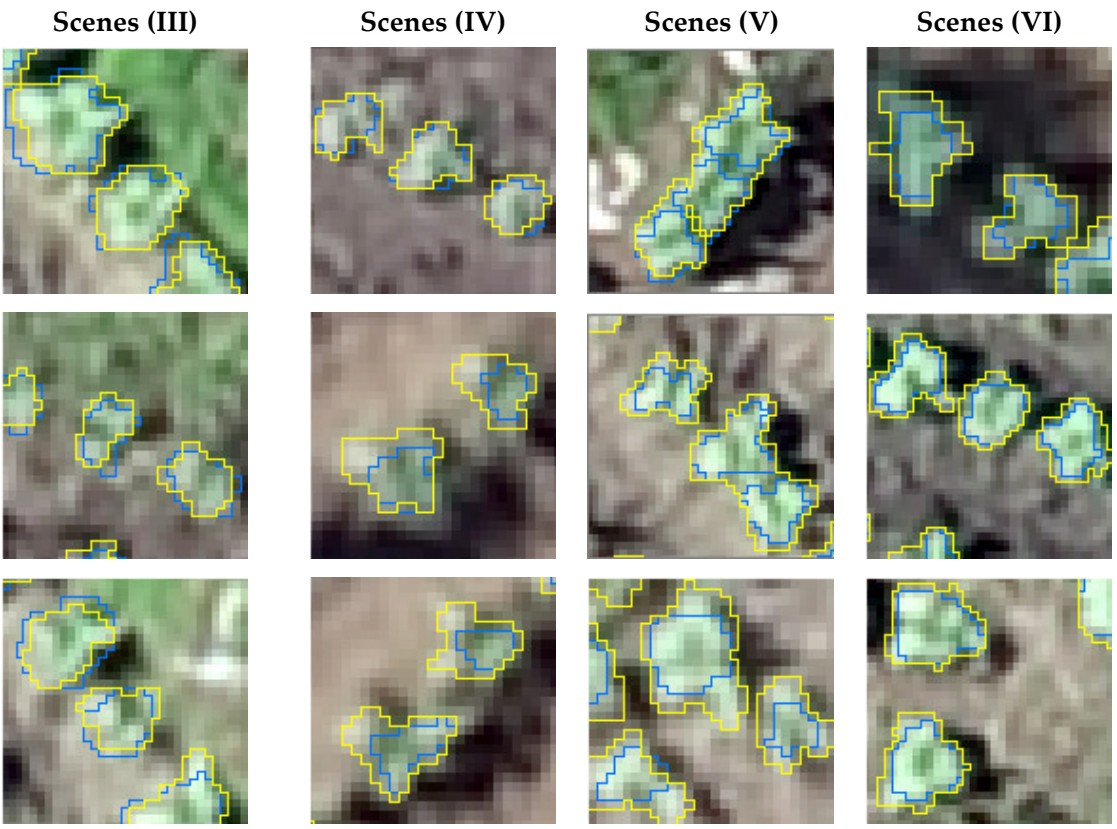

**Figure 10.** Recognition results of training the U-Net model with original and optimized samples (Note: blue color indicates original sample training results, and yellow color indicates optimized sample training results).

**Table 3.** Differences between the evaluation indexes of the original and optimized sample for training the U-Net model.

| Scenes | Precision | Recall | F1-Score | IOU |
|---|---|---|---|---|
| Smooth tectonics and weed-free (I) | 10.93% | 0.77% | 6.16% | 9.10% |
| Smooth tectonics and unevenly growing (II) | 3.82% | −9.35% | −2.08% | −2.87% |
| Smooth tectonics and weed-infested (III) | 25.31% | 7.68% | 18.10% | 20.34% |
| Smooth tectonics and planted with smaller seedlings (IV) | 22.81% | −7.81% | 8.92% | 10.74% |
| Subsurface fragmented and weed-free (V) | 0.78% | −4.17% | −1.80% | −2.65% |
| Surface fragmented and shadow-masked (VI) | 16.92% | −8.65% | 4.29% | 6.10% |
| Subsurface fragmented and weed-infested (VII) | 15.72% | −6.97% | 4.34% | 6.44% |
| Surface fragmented and planted with smaller seedlings (VIII) | 4.59% | −25.21% | −13.18% | −15.93% |

The optimization samples were optimized for the situation where the boundary feature information of the original tobacco. The weed samples were insignificant in the scene of fragmented surface and weeds. Thus, the boundary information of tobacco and weeds in the training sample can be more distinct. The generalization ability of the model can be enhanced, thus improving the model recognition accuracy. Tobacco planting had a time sequence. The late-planted tobacco plant was smaller. At the same flight altitude,

the feature information of smaller tobacco plants was not prominent, which affected the recognition accuracy. Therefore, the morphological features of the smaller tobacco plants in the training samples were optimized during the sample optimization. Then, the labeled plants can be closer to the real tobacco plants, improving the model recognition accuracy.

In summary, most models used in existing research have mainly been used for crops with relatively simple growth conditions and plant types, strong regularity of sowing spacing and plant types, relatively uniform spatial distribution of crop plants, relatively simple growing environmental conditions, and obvious and relatively homogeneous features of crop remote sensing images. In contrast, crops in karst mountainous areas have complex growing environments and structures. The crops show significantly dispersed three-dimensional spatial distribution and have nonuniform plant sizes. In this study, the U-Net model was used to identify tobacco plants in karst mountainous areas. The complex scene was deconstructed into eight single scenes. The accuracy of individual scenes and the overall accuracy were discussed separately. The accuracy was improved by optimizing the geometric features of the samples and other methods. The results in Table 1 indicate that the U-Net model has limitations in recognizing tobacco plants in complex habitats in karst mountainous areas.

## 5. Conclusions

The sample datasets derived from UAV visible light images were used to train the U-Net model. The experimental results indicate that it is effective and practical to some extent to use the trained model to identify tobacco plants in UAV visible light images under different planting environments. Scene segmentation reduced the interference of factors on the accuracy of tobacco plant identification, such as plot complexity and planting structure. This is a new attempt to improve the classification of crop recognition in complex habitats in karst mountainous areas (particularly habitats with fragmented surfaces).

The findings also reveal that the U-Net model showed different abilities in identifying features in different habitats due to the influence of some main factors such as plot fragmentation, plant size, presence of weeds, and shadow masking. Thus, it is necessary to construct the datasets by scene, increase samples, and eliminate interferences in a targeted manner according to the complexity of different scenes and the main factors affecting the model in order to improve the accuracy of the model in classifying complex scenes.

It was found that tobacco plant contour was the most significant influencing factor of the U-Net model in identifying tobacco plants in complex habitats in karst mountainous areas. This is related to the sample preparation error, followed by the accompanying interference of maize and weed. Maize, weeds, and tobacco are all green vegetation and have similar shapes, spectra, and texture features, leading to the misidentification of the U-Net model. In order to address this problem, the next step of this study is to increase the image bands and spectral information of the ground and use "shape-spectrum" joint features to eliminate different spectra or to improve the identification accuracy by removing the influence of noise, such as weeds, through morphological erosion and dilation operation.

The generalization ability and robustness of the U-Net model were strongly influenced by sample quality and quantity. The optimized sample dataset was used to train the model, which improved the sample profile, quality, and quantity. The accuracy of each scene was higher than the original sample. Therefore, in future research, we can further improve sample quantity and quality to improve the model performance.

**Author Contributions:** All authors contributed to the manuscript. Conceptualization, Y.H. and L.Y.; methodology, Q.L.; validation, D.H. and Y.H.; formal analysis, Q.L.; data curation, Y.H. and D.H.; writing—original draft preparation, L.Y.; writing—review and editing, Y.H., F.Z., D.H., L.C. and Q.L.; visualization, Y.H. and L.Y.; project administration, Z.Z.; funding acquisition, Z.Z. All authors have read and agreed to the published version of the manuscript.

**Funding:** This work was supported by the Guizhou Provincial Basic Research Program (Natural Science) ([2021] General 194) and Guizhou Provincial Key Technology R&D Program ([2023] General 211 and [2023] General 218), the Science and Technology program of Guizhou Province (Qiankehe Zhongyindi [2023]005).

**Institutional Review Board Statement:** Not applicable.

**Data Availability Statement:** All the data used in this study are mentioned in Section 2, "Materials and Methods".

**Acknowledgments:** The authors gratefully acknowledge the financial support of Guizhou Normal University. We would also like to thank the editors and anonymous reviewers for their helpful and productive comments on the manuscript.

**Conflicts of Interest:** The authors declare no conflicts of interest. The funders had no role in the design of the study, in the collection, analyses, or interpretation of data, in the writing of the manuscript, or in the decision to publish the results.

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
