# Peer review of "Complex Habitat Deconstruction and Low-Altitude Remote Sensing Recognition of Tobacco Cultivation on Karst Mountainous"

_agriculture, doi:10.3390/agriculture14030411_

Round 1

Reviewer 1 Report

Comments and Suggestions for Authors

Review of the manuscript entitled "Complex Habitats Deconstruction and Low Altitude Remote Sensing Recognition of Tobacco Cultivation on Karst Mountainous" (ID: agriculture-2846899).

·       In this manuscript, a U-Net model was trained to extract tobacco plant information from UAV visible images. An evaluation of the U-Net model for tobacco plant identification in complex habitats in the Karst mountains is presented here.

Despite the fact that the authors have put considerable effort into the paper, it needs to be better organized to be accepted. In my opinion, there are several points in the manuscript that need to be modified. Please see my comments and suggestions below:

- Page 3, Line 111-116; Page 4, Line 181-184: " A few references are needed in these sentences.

- Page 3, Line 116: The first time you use an acronym, write the phrase in full and place the acronym in parentheses immediately after it. You can then use the acronym throughout the rest of the text. For example, Convolutional Neural Network (CNN), Fully Convolutional Network (FCN), and ....

- Perhaps more importantly, what's new in this study compared to recent studies [1-3]? Please discuss it. The authors should indicate previous studies' weaknesses. This study's major innovation is unclear. Clearly explain the major innovation, and provide a relevant literature review that shows why your study is possibly a modeling innovation. In the introduction, it would be helpful to provide more details.

[1] He, L., Liao, K., Li, Y., Li, B., Zhang, J., Wang, Y., ... & Fu, X. (2024). Extraction of Tobacco Planting Information Based on UAV High-Resolution Remote Sensing Images. Remote Sensing, 16(2), 359.

[2] Xie, H., Zhang, W., Wu, Q., Zhang, T., Zhou, C., & Chen, Z. (2024). Extraction of tobacco planting area by using time-series of remote sensing data and the random forest algorithm. Remote Sensing Letters, 15(1), 24-34.

[3] Huang, L., Wu, X., Peng, Q., & Yu, X. (2021). Depth semantic segmentation of tobacco planting areas from unmanned aerial vehicle remote sensing images in plateau mountains. Journal of Spectroscopy, 2021, 1-14.

- Page 4, Line 159-160: "The image was acquired between 15:00-16:00 on June 4, 2021 (root extension stage), under clear weather and wind force 2.5, meeting the requirements for safe UAV operation." In this study, the focus was only on one date and one stage of tobacco plant growth. There may be differences in the biological and environmental conditions during the plant growth season and different stages, which may have an effect on the accuracy of the proposed method. Please explain why tobacco plant identification was not done at other dates and growth stages.

- Page 4, 2.3.1 U-Net model: In recent years, several variations of U-Net have been proposed to address common image segmentation challenges. Each one of these U-Net-related networks has advantages and disadvantages. ​ In this section, it is better to mention the disadvantages of the U-Net model.

- Page 5, Line 186: There is a noticeable similarity between Figure 2 with Figure 6 in [1]. At least the authors should pay attention to referencing (e.g., adapted from [1]).

[1] Zhou, Z., Peng, R., Li, R., Li, Y., Huang, D., & Zhu, M. (2023). Remote Sensing Identification and Rapid Yield Estimation of Pitaya Plants in Different Karst Mountainous Complex Habitats. Agriculture, 13(9), 1742.

- The quality of some figures should be improved. Missing Scale, North Arrow and Grid (coordinate system) in some figures, for example: Figure 1b and Figure 6 (size).

- The other major concern in the manuscript is the effective discussion of the results. It would be helpful if you could include a wide range of sources and put them in context with your own research. Within the discussion, include a detailed comparison of the findings of previous attempts. This should encompass the latest state-of-the-art approaches and validate this research's findings. Are there any implications and limitations to this research? Acknowledge and discuss any limitations or potential drawbacks of the proposed model. This can contribute to a more balanced presentation of the research.

Summary

Overall, the research is interesting and shows potential to be published in the journal. It is, however, not satisfactory in its current form due to several points that need improvement. I would suggest a major revision before the paper is accepted.

Comments on the Quality of English Language

The manuscript language is not clear enough and many places require more concise alternatives. Some sentences are complex and could be simplified for better readability. The manuscript has some minor errors. Some of these are mentioned in the attached PDF file. Corrections should be made throughout the manuscript.

Reviewer 2 Report

Comments and Suggestions for Authors

Hi Authors 

After a review of your document, here are some of the issues that need to be addressed

1. Ensure that all abreviations are written in full on first mention eg UAV

2. There are many words in your documents that are divided by a hyphen and this must be corrected.  line 54, 56, 57 and many others

3. Sentence 41 to 44 is too long and must be referenced. 

4. There is a lot of information which is coming from literature but is not not referenced. Can you ensure that you acknowledge all the sources where you got these facts. I noticed this on many occasion where a fact is coming from literature but you didnt cite the source. linr 48-50, 68-69, 80-81, 85-90, 98-107, 111-121, 191-194, 208-209, 217-219, 229-232, 

5. increase the font size of figure 1 and 2

6. can you write .shp in full for the benefit of non-gis readers. Either you use shapefile or vector

7. Line 252 - which arcmap tool did you use

8. In your results you mixed what you did (methods) with your outcomes. Can you please move what you did to the methodology section. This is common from line 358 to 391

9. line 326-328 must be moved to the discussion

10. line 355 to 356 - are these your findings or just facts from literature

11. Can you please benchmark your discusion with existing literature

11. 

Comments on the Quality of English Language

The language is fine

Round 2

Reviewer 1 Report

Comments and Suggestions for Authors

The current manuscript version could be considered for publication.
